# Ecological Connectivity in Two Ancient Lakes: Impact Upon Planktonic Cyanobacteria and Water Quality

**Matina Katsiapi** [1,2,*], **Savvas Genitsaris** [3], **Natassa Stefanidou** [1], **Anastasia Tsavdaridou** [4], **Irakleia Giannopoulou** [5], **Georgia Stamou** [5], **Evangelia Michaloudi** [5], **Antonios D. Mazaris** [4] and **Maria Moustaka-Gouni** [1,*]

[1]  Department of Botany, School of Biology, Aristotle University of Thessaloniki, 54124 Thessaloniki, Greece; natasa.stefanidou@gmail.com
[2]  EYATH SA, Water Supply Division/Drinking Water Treatment Facility, 57008 Nea Ionia, Greece
[3]  School of Science and Technology, International Hellenic University, 57001 Thermi, Greece; s.genitsaris@ihu.edu.gr
[4]  Department of Ecology, School of Biology, Aristotle University of Thessaloniki, 54124 Thessaloniki, Greece; atsavda@bio.auth.gr (A.T.); amazaris@bio.auth.gr (A.D.M.)
[5]  Department of Zoology, School of Biology, Aristotle University of Thessaloniki, 54124 Thessaloniki, Greece; giannopoulou.ira@gmail.com (I.G.); gstamouc@bio.auth.gr (G.S.); tholi@bio.auth.gr (E.M.)
*   Correspondence: matinakatsiapi@gmail.com (M.K.); mmustaka@bio.auth.gr (M.M.-G.)

**Abstract:** The ancient lakes Mikri Prespa and Megali Prespa are located in SE Europe at the transnational triangle and are globally recognized for their ecological significance. They host hundreds of flora and fauna species, and numerous types of habitat of conservational interest. They also provide a variety of ecosystem services. Over the last few decades, the two lakes have been interconnected through a surface water channel. In an attempt to explore whether such a management practice might alter the ecological properties of the two lakes, we investigated a series of community metrics for phytoplankton by emphasizing cyanobacteria. Our results demonstrate that the cyanobacterial metacommunity structure was affected by directional hydrological connectivity and high dispersal rates, and to a lesser extent, by cyanobacterial species sorting. Cyanobacterial alpha diversity was twofold in the shallow upstream Lake Mikri Prespa (Simpson index average value: 0.70) in comparison to downstream Lake Megali Prespa (Simpson index average value: 0.37). The cyanobacterial assemblage of the latter was only a strict subset of that in Mikri Prespa. Similarly, beta diversity components clearly showed a homogenization of cyanobacteria, supporting the hypothesis that water flow enhances fluvial translocation of potentially toxic and bloom-forming cyanobacteria. Degrading of the water quality in the Lake Megali Prespa in anticipation of improving that of the Lake Mikri Prespa is an issue of great concern for the Prespa lakes' protection and conservation.

**Keywords:** man-made surface water channel; transboundary; nestedness; Balkan; *Dolichospermum lemmermannii*; *Microcystis aeruginosa*

## 1. Introduction

One can view lakes as islands in a terrestrial world. In lakes connected to each other by direct water flow, the physical transport through water is unidirectional, resulting in a dominance of colonization from the upstream lake [1]. This way, the drift of phytoplankton, and particularly of cyanobacteria (both the resting stages and the individuals of active populations), are very effective means of dispersal and are not accidental episodes [2]. For cyanobacteria, high dispersal levels and global warming are emergent drivers of their community assembly in lakes [3]. In a lake that is already populated

by a given phytoplankton species, competition and predation by zooplankton makes it difficult for a new invader to establish. However, this "priority effect" sensu De Meester et al. [4] does not hold in cases where local communities are not saturated [5]. Recent invasions and proliferation of toxic cyanobacteria in diverse aquatic habitats, a well-known worldwide phenomenon [6], shows that new invaders can establish in a populated lake. Studying the species' dispersal patterns, recognising species' replacements (i.e., turnover) and recognising losses/additions of species in an ecosystem (nestedness) [7], could serve as critical steps towards implementing management practices [8].

The increase and dominance of cyanobacteria in a water body is indicative of water quality degradation, since cyanobacteria are implicated in food-web disturbances, oxygen depletion and animal mortality; and they have adverse effects on human health and on ecosystem services in general [9]. Furthermore, cyanobacterial species with allelopathic characteristics can alter phytoplankton composition and biodiversity [10]. It is, therefore, globally acknowledged that the management of lakes should aim at maintaining environmental heterogeneity while preventing further eutrophication and expansion of toxic and allelopathic cyanobacteria. This management practice could favour the maintenance of high phytoplankton beta and gamma diversity [11].

The ancient neighboring lakes Megali Prespa and Mikri Prespa, SE Europe, are connected by a man-made surface water channel with temporal flow. Over the period of 1984–2011, the two lakes had an almost identical range in water isotope composition, reflecting their hydrological connection [12]. The inflow from Lake Mikri Prespa is about 9% of the water inflows into the Lake Megali Prespa [13]. Thus, the community properties and water quality of both lakes could be driven by the plankton species of the Lake Mikri Prespa [14]. Still, no study exists on the impacts of the inflows from the Lake Mikri Prespa on the cyanobacteria and phytoplankton assemblages and the water quality of the Lake Megali Prespa. Notwithstanding, the Lake Mikri Prespa has a history of cyanobacterial blooms formed by potentially toxic species [15,16]. In 2014, microcystins were measured in lake water at a concentration that posed a high risk of adverse human health effects [17].

In the present work, we examined past hydrological connectivity's cumulative effect on the Prespa lakes' cyanobacteria assemblages and the lake's water quality. Towards those ends, we studied the two lakes' phytoplankton communities over a year without water transfer, focusing on cyanobacteria. We examined temporal variation in composition, biomass, alpha diversity, total beta diversity and the species' turnover and nestedness as the means to delineate the potential role of hydrological connectivity upon the phytoplankton assemblages. Species network interactions within each lake were also explored as the means to identify co-occurrence patterns and negative interactions, while the multimetric index PhyCoI$_{GP}$ [18] was applied to compare water quality of the two lakes.

## 2. Materials and Methods

### 2.1. Study Site

The lakes Mikri Prespa and Megali Prespa are of two the oldest lakes on earth, both with an estimated age of 2–5 million years [19]. These two lakes are European endangered monuments hosting high richness of endemic and internationally important endangered species. The two lakes are also included in the Natura 2000 network that constitutes the cornerstone of the conservation policy of the European Union. Since 1975, Lake Mikri Prespa was also designated as a Ramsar Site. Both lakes are situated at the northwestern part of Greece, SE Europe (Megali Prespa: 40°45' N, 21°01' E, Mikri Prespa: 40°46' N, 21°05' E) at an altitude of ≈850 m above sea level (Figure 1). The Lake Mikri Prespa is also a transboundary lake between Greece and Albania. The Lake Megali Prespa is located at the southernmost tip of the Alpine biogeographical region of Europe [20], and is shared by three countries: Greece, Albania and North Macedonia.

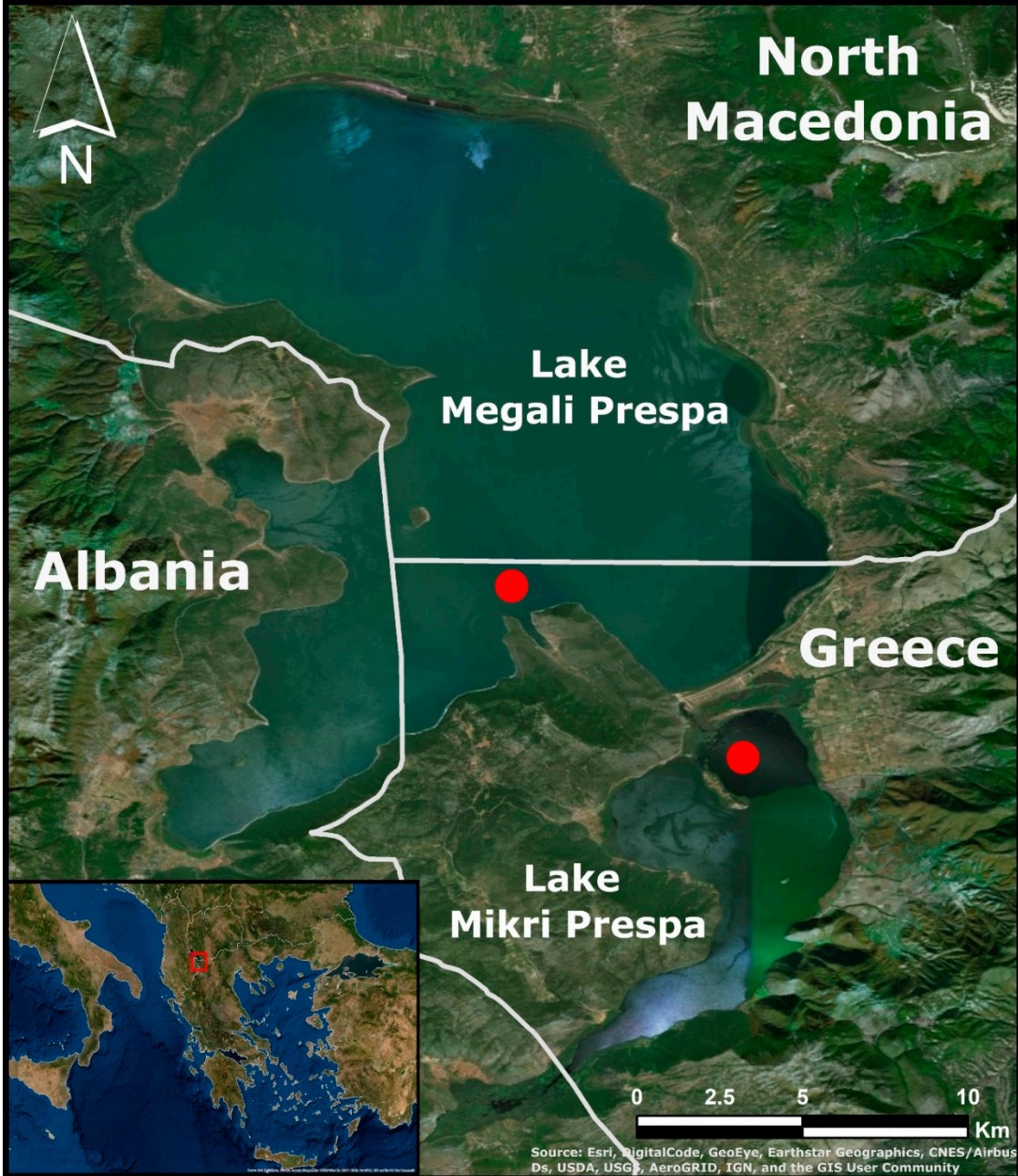

**Figure 1.** Study area and sampling sites, the lakes Mikri and Megali Prespa, indicated with red circles.

Lake Mikri Prespa is a shallow (maximum depth: 9 m), moderately large (surface: 46.7 km$^2$) polymictic, eutrophic lake experiencing toxic cyanobacterial blooms [17,21]. Lake Megali Prespa is a large (surface 256 km$^2$), deep (maximum depth 55 m), warm momomicitc lake, less impacted by humans [22,23]. The two lakes (Figure 1) are connected by a man-made channel which used to be narrow and quite shallow until the 1950s, but was re-constructed in 1969. Since 2005, the Prespa Park Management Body controls the water level by regulating the outflow discharge to the Lake Megali Prespa in order to sustain the Lake Mikri Prespa's good ecological status and the local economy [24]. During the twelve years of controlled discharge (2005–2016), a total of $212.6 \times 10^6$ m$^3$ of surface lake water of the Lake Mikri Prespa was discharged to the Lake Megali Prespa based on the water level adjustments [25].

### 2.2. Field Work—Microscopy Analysis

Phytoplankton samplings were conducted in the lakes Mikri and Megali Prespa, from September 2015 to August 2016, on a monthly basis. Water samples were collected at the deeper parts of the lakes in the vicinity of the connection points of the lakes to the water channel (Mikri Prespa outlet, Megali Prespa inlet; Figure 1) using a Niskin sampler. The samples were taken at discrete depths every one meter from the surface of the lakes to the bottom of the euphotic zone (defined as 2.5× Secchi Depth). The set of these samples in each sampling site were mixed in a plastic container, and an integrated sample from the euphotic zone was thus obtained. Subsamples of 500 mL for phytoplankton analysis were preserved immediately after sampling with Lugol's solution and were kept in the dark till microscopy analysis. Also, fresh (non-preserved) sub-samples were kept in the dark in a portable refrigerator. The fresh samples were transported immediately at the laboratory and checked (within 4–6 h from sampling) for the microscopic identification of species with limited diagnostic features.

The phytoplankton data were based on the microscopic analysis of the preserved samples. For this, sub-samples were placed in sedimentation chambers of Hydrobios of different volumes (3, 5, 10 and 25 mL) based on phytoplankton density according to the inverted microscope method. These were examined using an inverted microscope (Nikon Eclipse TE 2000-S, Melville, NY, USA). Species identification was carried using appropriate taxonomic keys [26]. Phytoplankton counting was performed using Utermöhl's sedimentation method. At least 400 individuals were counted in each sample. For biomass estimation, the dimensions of 30 individuals of dominant species were measured using the relevant tools of a digital microscope camera (Nikon DS-L1, Melville, NY, USA). Mean cell or filament volume estimates were calculated using appropriate geometric formulae [27].

### 2.3. Data Analysis

Alpha diversity estimators (i.e., species richness and the Simpson index) for the whole phytoplankton community and for cyanobacteria were calculated with the PAST3 software [28]. To compute beta diversity of the whole phytoplankton community and of the cyanobacteria assemblages in each sampling, we used the "betapart" R package, version 1.5.1 [29]. Beta diversity was portioned into its spatial turnover and nestedness components, following Baselga's approach [30]. This approach [7,30] suggests that Sorensen pair-wise dissimilarity (bSOR) should be partitioned into two components: spatial turnover in species composition, measured as Simpson pair-wise dissimilarity index (bSIM); and variation in species composition due to nestedness (bNES) measured as the nestedness-fraction of Sorensen pair-wise dissimilarity. The above analyses were run in the R 3.5.3 environment (R core team 2018).

We performed network analysis for each lake to explore significant relationships (positive, which indicate co-occurrence patterns; or negative, which provide evidence of exclusion) among cyanobacterial and other phytoplankton species. The relationships were characterized through Maximal Information-based Nonparametric Exploration (MINE) statistics by computing the Maximal Information Coefficient (MIC), based on the species biomass per sample, in species pairwise comparisons (see Supplementary Table S1 for the species included) [31]. The matrix of MIC values corresponding to a *p*-value < 0.01, based on pre-computed *p*-values of various MIC scores at different sample sizes, was used (MIC > 0.68 in this case) in order to visualize networks of species' associations with Cytoscape 3.5.1 [32]. We identified the negative or positive type of relationship between each pair of species included in the network according to Hernández-Ruiz et al. [33].

### 2.4. Water Quality Assessment

The phytoplankton modified PhyCoI$_{GP}$ index [18] was used to assess ecological water quality of each lake. For the index calculation, we combined six metrics/sub-indices; i.e., the total phytoplankton biomass, the cyanobacterial biomass (according to World Health Organization Guidelines), the modified

Nygaard sub-index based on the biomass of indicator taxonomic groups, the modified Nygaard sub-index based on species richness of indicator taxonomic groups, the quality group species Index and the grazing potential of zooplankton. The ecological water quality assessment was based on the data of the sampling period June–September. This period is used for the lake ecological water quality assessments based on phytoplankton in the Greece/Mediterranean region [18].

## 3. Results

In total, 119 species were identified in the phytoplankton communities of the Prespa lakes; 111 species were found in Lake Mikri Prespa and 75 phytoplankton species were identified in Lake Megali Prespa. Sixty-seven (67) phytoplankton species were found in both lakes (Supplementary Figure S1). Chlorophytes were the richest phytoplankton taxonomic group (47 species). A total of 30 species of Cyanobacteria were detected in both lakes (20 species in Lake Megali Prespa and 30 in Lake Mikri Prespa) (Supplementary Table S1). The cyanobacterial assemblage in Lake Megali Prespa was a strict subset of the cyanobacterial assemblage of Lake Mikri Prespa. All nostocalean species (*Dolichospermum* cf. *flos-aquae, D. lemmermannii, D. viguieri, Aphanizomenon gracile*) were found in both lakes. Most of the cyanobacteria that were not observed in the Lake Megali Prespa (eight out of 10) were chroococcalean; two of them were *Microcystis* species (*M. flos-aquae* and *M. wesenbergii*). The maximum cyanobacteria richness was detected in September and the minimum in February (Table 1).

**Table 1.** Total phytoplankton biomass (TB), cyanobacterial assemblage biomass (CB), species richness of phytoplankton community (SR-*ph*), species richness of cyanobacterial assemblage (SR-*cy*), Simpson index of phytoplankton community (SI-*ph*), Simpson index of cyanobacterial assemblage (SI-*cy*) and the modified phytoplankton water quality index (PhyCoI$_{GP}$) in the lakes Mikri and Megali Prespa during the period from September 2015 to August 2016. Values for Lake Megali Prespa are in bold.

| Date | TB (mg L$^{-1}$) | | CB (mg L$^{-1}$) | | SR-*ph* | | SR-*cy* | | SI-*ph* | | SI-*cy* | | PhyCoI$_{GP}$ | |
|---|---|---|---|---|---|---|---|---|---|---|---|---|---|---|
| September 2015 | 3.07 | **1.18** | 2.47 | **0.29** | 64 | **36** | 26 | **12** | 0.80 | **0.69** | 0.70 | **0.73** | 2.7 | **3.1** |
| October 2015 | 2.37 | **1.72** | 1.93 | **0.07** | 57 | **28** | 19 | **9** | 0.84 | **0.65** | 0.77 | **0.55** | 2.7 | **2.7** |
| November 2015 | 1.18 | **1.46** | 0.86 | **0.03** | 56 | **28** | 20 | **9** | 0.81 | **0.54** | 0.67 | **0.26** | 3.0 | **3.1** |
| December 2015 | 0.92 | **0.89** | 0.67 | **0.003** | 43 | **19** | 15 | **2** | 0.78 | **0.18** | - | - | 3.3 | **3.6** |
| January 2016 | 0.23 | **0.10** | 0.08 | **0.001** | 36 | **17** | 11 | **3** | 0.90 | **0.69** | - | - | 3.5 | **4.2** |
| February 2016 | 0.72 | **0.12** | 0.17 | **0.001** | 33 | **19** | 9 | **2** | 0.57 | **0.45** | - | - | 3.5 | **3.8** |
| March 2016 | 0.70 | **0.58** | 0.11 | **0.002** | 51 | **18** | 17 | **3** | 0.78 | **0.20** | - | - | 3.6 | **4.3** |
| April 2016 | 1.54 | **0.41** | 0.38 | **0.05** | 46 | **18** | 17 | **4** | 0.63 | **0.75** | 0.72 | **0.19** | 3.3 | **3.9** |
| June 2016 | 0.87 | **4.10** | 0.54 | **2.63** | 66 | **27** | 22 | **5** | 0.86 | **0.53** | 0.71 | **0.004** | 3.7 | **2.6** |
| July 2016 | 1.07 | **0.72** | 0.43 | **0.10** | 54 | **15** | 19 | **6** | 0.85 | **0.72** | 0.77 | **0.43** | 3.5 | **2.9** |
| August 2016 | 1.93 | **0.76** | 0.91 | **0.23** | 54 | **20** | 19 | **8** | 0.81 | **0.75** | 0.57 | **0.43** | 3.4 | **3.2** |

During the study period, the phytoplankton biomass ranged from 0.10 to 4.1 mg L$^{-1}$ in the two lakes (Figure 2). The highest value was recorded in Megali Prespa in June 2016, whereas in Lake Mikri Prespa the maximum biomass reached 3.07 mg L$^{-1}$ in September 2015. The cyanobacterial biomass reached 2.47 mg L$^{-1}$ in Lake Mikri Prespa and 2.63 mg L$^{-1}$ in Lake Megali Prespa. The highest annual contribution of cyanobacteria to phytoplankton biomass was observed in Lake Mikri Prespa (58.6%). We found that the cyanobacterium *Dolichospermum lemmermannii* was forming water blooms in both lakes, but in different periods, and contributed up to 23.1% of the annual cyanobacterial biomass in Lake Megali Prespa (Figure 2). The cyanobacterium *Microcystis aeruginosa,* was rarely detected in Lake Megali Prespa, but dominated the cyanobacterial biomass in Lake Mikri Prespa (34.6% annual contribution, which was followed by *D. lemmermannii* (14.0%) and *Microcystis panniformis* (5.4%),

another rarely recorded species in Lake Megali Prespa where the second species in terms of biomass was *Aphanizomenon gracile* (2.4% annual contribution).

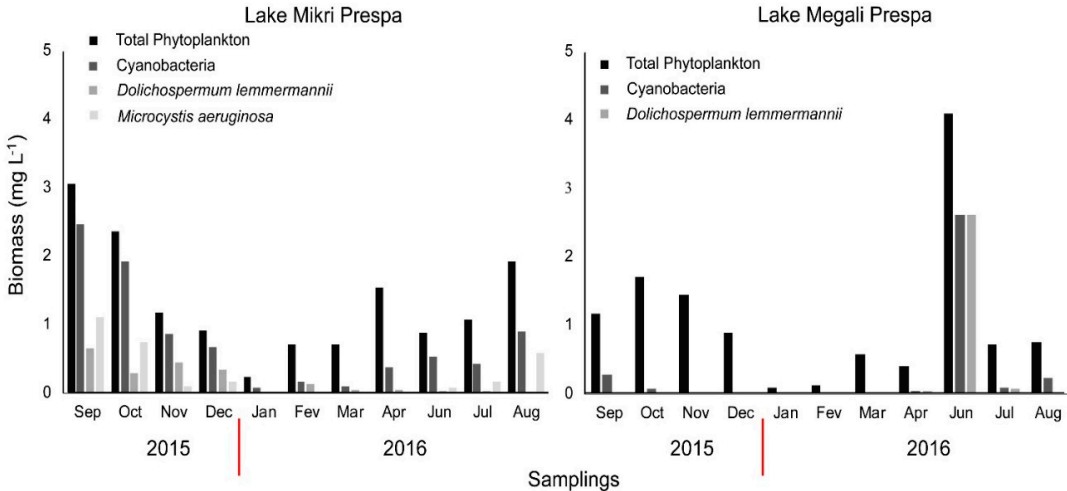

**Figure 2.** Total phytoplankton biomass, cyanobacterial assemblage biomass and *Dolichospermum lemmermannii* and *Microcystis aeruginosa* biomasses in the lakes Mikri and Megali Prespa during the period from September 2015 to August 2016. *M. aeruginosa* biomass was counted only in the samples of Lake Mikri Prespa.

Phytoplankton alpha diversity (Simpson index) varied between 0.18 and 0.90 in both lakes, and the cyanobacterial assemblage alpha diversity during their growth period (September–November 2015 and April–August 2016) varied between 0.004 and 0.73 in Lake Megali Prespa (average 0.37) and 0.57 and 0.77 (average 0.7) in Lake Mikri Prespa. Phytoplankton beta diversity (bSOR) ranged from 0.43 to 0.63 (Figure 3) and cyanobacterial beta diversity ranged from 0.38 to 0.7. Phytoplankton community compositional variation was attributed equally to the species turnover (50.1%) and nestedness (49.9%) while cyanobacterial variation during their growth period was mostly attributed to nestedness (77.4%) rather than species turnover (22.6%). For total phytoplankton, both components had substantial monthly fluctuations, while for cyanobacteria, nestedness was always higher (Figure 3).

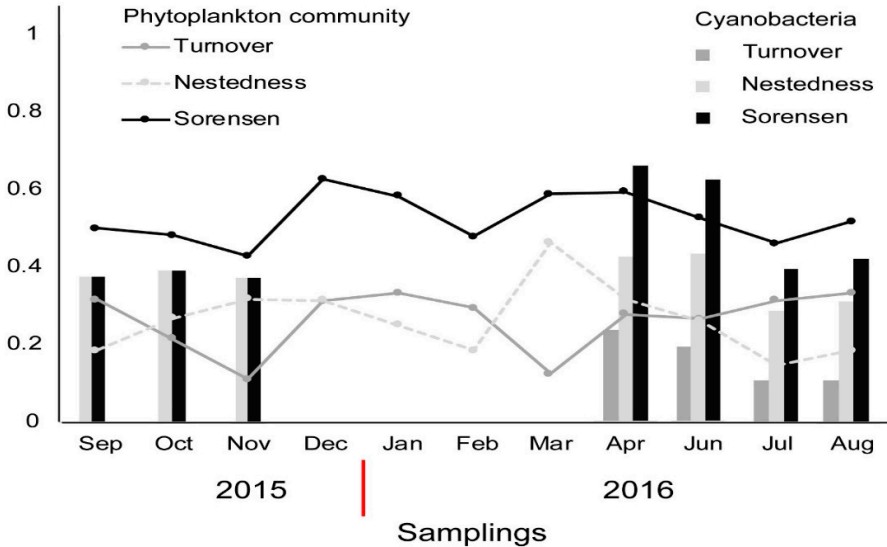

**Figure 3.** Phytoplankton and cyanobacterial assemblage beta diversity (bSOR), and their turnover and (bSIM) and nestedness (bNES) components values in the lakes Mikri and Megali Prespa.

The strong connections between cyanobacterial/phytoplankton species according to MIC correlation coefficients (MIC values corresponding to *p*-values < 0.01) are visualized in networks (Figure 4). Several negative and positive connections between cyanobacteria and other phytoplankton taxa were observed. The observed links were attributed to Cyanobacteria and Chlorophyta. In particular, the dominant cyanobacteria in Lake Mikri Prespa, *M. aeruginosa* and *D. lemmermannii* were negatively connected. In Lake Megali Prespa, *D. lemmermannii* had negative connections both with the other nostocalean species *Dolichospermum viguieri* and *Aphanizomenon gracile* and the oscillatorealean *Planktolyngbya limnetica*.

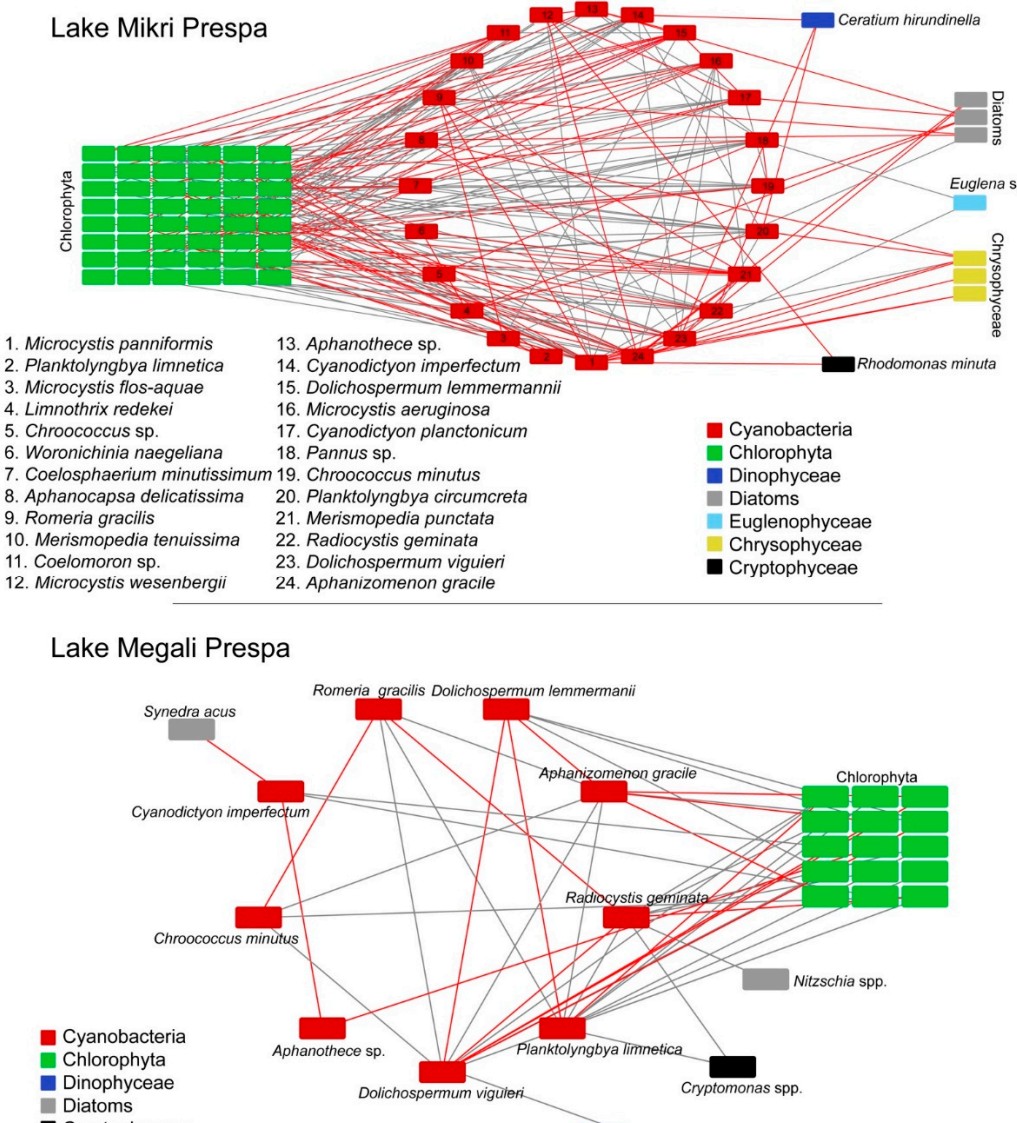

**Figure 4.** Network diagrams of the phytoplankton species (nodes) showing correlations (edges) with highly significant connections (*p*-values < 0.01) based on the maximal information coefficient (MIC) scores in the lakes Mikri and Megali Prespa. A positive relationship between each pair of phytoplankton species is depicted with grey edges while negative with red edges. Different node colors represent different taxonomic groups according to the color key.

The values of the water quality index PhyCoI$_{GP}$ ranged from 2.6 to 4.3 in Lake Megali Prespa and from 2.7 to 3.7 in Lake Mikri Prespa (Table 1). Ecological water quality assessment indicated a similar level for both lakes; i.e., around the good–moderate boundary. In particular, Lake Megali Prespa was

classified in moderate water quality (average index value 2.9) and Lake Mikri Prespa was classified in the good class (average index value 3.3).

## 4. Discussion

Our focus on cyanobacterial species spatial distribution and community properties in two hydrologically connected ancient European lakes was motivated by the scarce information on hydrological connectivity as an emergent driver for cyanobacterial species homogenization in global expansion studies of toxic and non-toxic species [3,34]. The species pool of phytoplankton communities was found different in the two lakes with the average value of Simpson index almost double in the shallow, eutrophic Lake Mikri Prespa [21]. This result was rather expected because of the different typology (i.e., depth, surface area, stratification type) of the two lakes as well as their different trophic states, all significant determinants of phytoplankton alpha diversity [22,35]. However, we detected a high similarity in cyanobacterial diversity between these two different lakes: the cyanobacterial species of Lake Megali Prespa were a strict subset of the species pool inhabiting the shallow Lake Mikri Prespa. The increased similarity in cyanobacterial composition could be explained, to some extent, by the high occurrence of nostocalean species, bearing akinetes, cyanobacteria. These dormant cells survive long and extreme dispersion routes, assuring perennial germination and proliferation of toxic and non-toxic blooms [36].

The spread of nostocalean species from Lake Mikri Prespa to Lake Megali Prespa can be supported by the fluvial immense seeds of their cells (dormant and vegetative). Planktic nostocalean cyanobacteria have the ability to produce most types of cyanotoxins [37] and are frequently referred to as successful invaders [6]. *Dolichospermum lemmermannii*, the dominant nostocalean species in both lakes in terms of biomass, is known for its ability to produce various toxins (e.g., microcystins, anatoxins, saxitoxin) [38]. Also, *D. lemmermannii* has widened its geographic distribution across temperate lakes. Moreover, its ability to fix nitrogen underlies its competitive advantages in a lake with low inorganic nitrogen [39]. It has been reported to flourish in Lake Mikri Prespa since 1990 [15]. In this lake, in 2014, when *D. lemmermannii* and *Microcystis aeruginosa* formed a bloom, microcystins were measured at a concentration that posed a high risk of adverse human health effects [17]. Successful dispersal of nostocaleans could be favoured by their large population sizes reaching bloom densities [40].

However, all cyanobacteria species were not nostocalean nor identical in both lakes. The differences in cyanobacterial composition could be explained to some extent by the lakes' environmental filtering and species traits. For example, the other bloom-forming toxic cyanobacterium in Lake Mikri Prespa, *M. aeruginosa*, was found in low numbers in Lake Megali Prespa, whereas other *Microcystis* species (*M. flos-aquae* and *M. wesenbergii*) were not detected. *M. aeruginosa* is differentiated from the co-existing nostocalean species by its preference for higher phosphorus concentrations and for higher mixing depth/euphotic depth and nitrogen/light conditions ratios in the neighboring Lake Kastoria [41]. It seems that the lower trophic state and different lake typology characteristics are the environmental barriers for the *Microcystis* population increase in Lake Megali Prespa. The negative connection between *M. aeruginosa* and *D. lemmermannii*, shown in the network of Lake Mikri Prespa, may suggest that these species have temporally separated niches, indicating different species traits and preferences [1]. Lake environmental filtering and cyanobacterial species traits (dispersal abilities and competitive advantages), as described above, may play an important role in determining the Prespa lakes' metacommunity structure [42]. Geographic expansion of chroococcalean cyanobacteria has been frequently reported, as intensive blooms of toxic *Microcystis* have been attributed to global warming and regional eutrophication [6]. In addition, the influence of hydrological connectivity on toxic *Microcystis* spreading between two drowned river mouth lakes resulted in *Microcystis* dominance in both lakes [43]. Nevertheless, local conditions in each lake were important in determining which cyanobacterial species could maintain a viable population. The other chroococcalean species did not co-occur in the Prespa lakes. This might indicate different environmental preferences and habitat checkboards [44].

Phytoplankton, and particularly, cyanobacteria, are subjected to directional dispersal (water flow), a regional factor leading different habitat communities to be highly connected [45]. Thus, habitat connectivity and dispersal interact to structure metacommunity. The cyanobacterial dispersal because of the directional water flow could potentially override habitat control leading to species occupying unfavourable habitats [46]. Our analyses demonstrated a moderate to low dissimilarity in phytoplankton and cyanobacteria in the Prespa lakes. This is in contrast with results from other lakes and reservoirs of the world with strong environmental heterogeneity [11,47]. High beta diversity in these freshwaters was mainly explained by species turnover. In the Prespa lakes, beta diversity was explained equally by the species turnover and nestedness, while cyanobacterial assemblage beta diversity was mainly attributed to nestedness. In an analysis of freshwater plankton, bacteria were found to have a higher degree of nestedness than phytoplankton did, whereas the degree of nestedness was related mainly to habitat quality [44]. A meta-analysis of nestedness and turnover components of beta diversity across organisms and ecosystems by Soininen et al. [48] showed that passively dispersed organisms (e.g., phytoplankton) had lower turnover and total beta diversity than flying organisms but still much higher values than in the Prespa lakes.

In the Prespa lakes, different typologies and trophic states would initially lead to an assumption of low cyanobacterial nestedness. However, long-term hydrological connectivity of the two lakes appears to override habitat control through source-sink dynamics, leading to cyanobacterial species also occupying unfavourable habitats [46]. Furthermore, the higher cyanobacteria nestedness in relation to other phytoplankton taxa found in our study might also show the cyanobacteria's ability to adapt to unfavourable conditions. Kraus et al. [49] studying hydrological variations and ecological phytoplankton patterns in Amazonian floodplain lakes, explained the absence of phytoplankton dissimilarities to the ability of cyanobacteria to adapt to contrasted ecological conditions. No dispersal limitation and high levels of stochastic cyanobacterial establishment with a weak influence of phosphorus has been found in peri-Alpine lakes [3]. Cyanobacterial biodiversity patterns over one century of eutrophication and climate change showed clear signs of beta diversity loss with homogenization across sites in recent decades. This suggests that potentially toxic and bloom-forming cyanobacteria can widen their geographic distribution in the European temperate region [3]. Monchamp et al. [3] also found that *Dolichospermum* species were able to colonize a wide range of lakes. *Dolichospermum* species' distribution in the Prespa lakes agrees with these findings. The expansion of the bloom-forming, potentially toxic *D. lemmermannii* in several deep southern subalpine lakes has also been recently reported [40]. The species is considered as one of the most problematic algae in the subalpine Lake District, raising serious concerns because of the impacts on the tourist economy and the potential toxigenic effects. There is also an interesting expansion of this species between the 40th parallel and the Arctic Circle, which is in contrast to the prevailing south to north dispersion paths typical of other Nostocales [40]. In this case, our study provides important information about *D. lemmermannii*'s southernmost distribution in a deep and large lake (i.e., Lake Megali Prespa).

Based on the results of the application of the PhyCoI$_{GP}$ index for Mediterranean lakes, the water quality of Lake Megali Prespa cannot be considered good. The water quality of Lake Mikri Prespa showed considerable improvement characterized by much lower total phytoplankton and cyanobacterial biomas than during the period of 1990–2010 [15,22]. Moreover, based on phytoplankton biomass data of the period 2012–2014 and using the Alpine classification system for lake types L-AL3 and L-AL4, Lake Megali Prespa was also classified at lower than good ecological class [22]. The use of Alpine classification system was based on the location of Lake Megali Prespa at the southernmost tip of the Alpine biogeographical region of Europe [20].

We conclude that the Prespa lakes' cyanobacterial metacommunity structure was mainly affected by directional hydrological connectivity and high dispersal rates (Mikri Prespa source—Megali Prespa sink), a mass effect paradigm. Furthermore, it was affected to a less extent by species sorting (mostly for chroococcalean species) reflected by low beta diversity and high nestedness.

The hydrological connectivity is an emergent driver for cyanobacterial species homogenization, water quality homogenization and potentially toxic bloom expansion. The degrading of the water quality in Lake Megali Prespa in anticipation of improving that of Lake Mikri Prespa is an issue of great concern for lakes management. Our study supports the fact that the management of the Prespa lakes should aim at maintaining environmental heterogeneity while preventing the species of Lake Mikri Prespa, which include potentially toxic cyanobacteria, flooding into Lake Megali Prespa.

**Supplementary Materials:** The following are available online at http://www.mdpi.com/2073-4441/12/1/18/s1. Figure S1: Venn diagram showing the number of unique and common phytoplankton and cyanobacterial species between the lakes Mikri and Megali Prespa. Table S1: Phytoplankton species list that was recorded from the lakes Mikri and Megali Prespa during the period September 2015 to August 2016. X indicates presence; $ indicates participation in the networks.

**Author Contributions:** Conceptualization, M.K., A.D.M. and M.M.-G.; data curation, M.K., S.G. and N.S.; resources, E.M. and M.M.-G.; writing—original draft, M.K., S.G., N.S., A.D.M. and M.M.-G.; writing—review and editing, M.K., S.G., N.S., A.T., E.M., A.D.M. and M.M.-G.; visualization, M.K., S.G., N.S., A.T., I.G. and G.S.; supervision, A.D.M., E.M. and M.M.-G.; project administration, A.D.M. and M.M.-G.; funding acquisition, A.D.M., E.M. and M.M.-G. All authors have read and agreed to the published version of the manuscript.

**Funding:** This research was funded by ProLife, a project co-funded by the European Union and by national funds of the participating countries under the IPA Cross-Border Programme "Greece—The Republic of North Macedonia 2007–2013." The views expressed in this publication do not necessarily reflect the views of the European Union, the participating countries and the Managing Authority.

**Conflicts of Interest:** The authors declare no conflict of interest.

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
