# Peer review of "Ecological Connectivity in Two Ancient Lakes: Impact Upon Planktonic Cyanobacteria and Water Quality"

_water, doi:10.3390/w12010018_

Round 1

Reviewer 1 Report

After minor revision (complete English correction and misspelling) paper can be accepted.

Mispellings like:

e.g. Row 30. Nestdeness---->nestedness

row 74. was applied to compare water quality of the  two lakes (there is no end of the sentences) and so on.

Author Response

Reviewer 1

Comments and Suggestions for Authors

After minor revision (complete English correction and misspelling) paper can be accepted.

We thank the reviewer for his/her positive disposition towards our manuscript. The manuscript has been proofread carefully and English corrections of misspellings have been made. All changes are highlighted in red in the revised version of the manuscript to facilitate reading.

Mispellings like:

e.g. Row 30. Nestdeness---->nestedness.

Corrected.

row 74. was applied to compare water quality of the  two lakes (there is no end of the sentences) and so on.

Corrected.

Reviewer 2 Report

The manuscript by Katsiapi et al. explores how the interconnection between the two ancient lakes as a part of management practice led to the changes in cyanobacterial community structure and water quality. The authors have done a decent job in conducting this study and will pave way for framing environmental policies for better management of freshwater systems. Although the subject is interesting, the manuscript is let down by its poor writing at certain places. At some places, sentences are either too long or very complicated to understand. The sentences can be made simple so that it is understandable for the general readers.

Specific comments

L14: should be “numerous”

L22-25: please rephrase. Instead of biomass between the two lakes which is more of less similar, why can’t the authors provide the alpha diversity value which is twofold higher in Mikri than Megali. The use of Prespa for both the lakes brings in some sort of confusion throughout the manuscript. Can they just stick to Lake Mikri and Megali?

L39: should be “In a lake that is already…

L40: should be “makes”

L44-47: Hard to understand. Rephrase it.

Information on samples collection such as sampler used, how depth integrated samples were obtained, transport time from field to laboratory, quantity of samples each analysis, whether the samples were protected from heat and light radiation during transportation?

L148-149: needs clarity.

Fig.2: in my opinion can be moved to supplementary material as it is described in the text.

L168: here why 4.1mg l-1 again? Already provided in the range (previous sentence).

Fig.4: Why values (Dec-March) for cyanobacteria are missing?

L220-223: Very hard to follow. Split up and make it easier to understand.

Author Response

Reviewer 2

The manuscript by Katsiapi et al. explores how the interconnection between the two ancient lakes as a part of management practice led to the changes in cyanobacterial community structure and water quality. The authors have done a decent job in conducting this study and will pave way for framing environmental policies for better management of freshwater systems. Although the subject is interesting, the manuscript is let down by its poor writing at certain places. At some places, sentences are either too long or very complicated to understand. The sentences can be made simple so that it is understandable for the general readers.

We thank the reviewer for his/her positive disposition towards our manuscript. The manuscript has been proofread carefully and changes in the text have been made to improve it according to the reviewer’s suggestions. We hope that the text is now more clear and easier to follow. All changes are highlighted in red in the revised version of the manuscript to facilitate reading.

Specific comments

L14: should be “numerous”

Corrected.

L22-25: please rephrase. Instead of biomass between the two lakes which is more of less similar, why can’t the authors provide the alpha diversity value which is twofold higher in Mikri than Megali.

Thank you for the suggestion. We now provide the average Simpson index value of the cyanobacterial assemblages during their growth period in each of the two lakes.

The use of Prespa for both the lakes brings in some sort of confusion throughout the manuscript. Can they just stick to Lake Mikri and Megali?

We understand that this repetition might be tiring and confusing in some parts. Unfortunately, the lakes’ names are Megali Prespa, and Mikri Prespa, so it is not possible to change this.

L39: should be “In a lake that is already…

Thank you, we have now changed the text according to the suggestion.

L40: should be “makes”

Corrected.

L44-47: Hard to understand. Rephrase it.

This part is now re-written.

Information on samples collection such as sampler used, how depth integrated samples were obtained, transport time from field to laboratory, quantity of samples each analysis, whether the samples were protected from heat and light radiation during transportation?

Thank you. All this missing information is now added in the revised text in L103-115.

L148-149: needs clarity.

Thank you, we have now rephrased this part.

Fig.2: in my opinion can be moved to supplementary material as it is described in the text.

Figure 2 is now moved to supplementary material, as Supplementary Figure S1. All figure numberings have been changed in text accordingly.

L168: here why 4.1mg l-1 again? Already provided in the range (previous sentence).

Thank you. We have now deleted the parenthesis.

Fig.4: Why values (Dec-March) for cyanobacteria are missing?

Cyanobacterial growth terminated in both lakes in November and restarted in April, particularly in the Lake Megali Prespa as it can be seen from the very low values of their biomass during December – March (Table 1). Even in the Lake Mikri Prespa the cyanobacterial biomass during this period was the residual of previous growth.  For our analysis we needed to exclude the possibility that the cold period biomass was just a non-growing residual from the previous autumn. Taxonomic shifts from autumn to winter also indicated no active growth of cyanobacteria in winter. Based on this, we examined cyanobacterial diversity for their growth period that is September – November 2015 and April – August 2016.

L220-223: Very hard to follow. Split up and make it easier to understand.

Thank you. We have now modified the sentence according to the suggestion.

Round 2

Reviewer 2 Report

All my concerns are sufficiently addressed in the revised version of the manuscript.